# Mapping the Gut Microbiota Composition in the Context of Raltegravir, Dolutegravir, and Bictegravir—A Scoping Review

**DOI:** 10.3390/ijms26136366

**Published:** 2025-07-02

**Authors:** Zsófia Gáspár, Botond Lakatos

**Affiliations:** 1National Institute of Haematology and Infectious Diseases, Central Hospital of Southern Pest, H-1097 Budapest, Hungary; botond.lakatos@dpckorhaz.hu; 2Doctoral School of Clinical Medicine, Semmelweis University, H-1085 Budapest, Hungary; 3Departmental Group of Infectious Diseases, Department of Internal Medicine and Hematology, Semmelweis University, H-1088 Budapest, Hungary

**Keywords:** dolutegravir, bictegravir, raltegravir, HIV, microbiome, gut

## Abstract

(1) Background: Second-generation integrase strand transfer inhibitors (INSTIs) are now the preferred first-line therapies for human immunodeficiency virus (HIV). However, concerns regarding their side effects, such as weight gain and metabolic disturbances, have emerged. This scoping review aims to assess the effects of INSTIs on the gut microbiota, with a focus on differences between agents and their clinical implications. (2) Methods: A scoping review was conducted using PubMed, Web of Science, and Embase, with reports collected following PRISMA for Scoping Reviews (PRISMA-ScR). (3) Results: The majority of available evidence focused on dolutegravir, which demonstrated beneficial effects on microbiota diversity and composition. However, factors such as younger age, lower CD4+ counts, and extreme BMI were associated with proinflammatory changes. Limited data on bictegravir also suggested favorable alterations in the gut microbiota. Raltegravir, a first-generation INSTI, was associated with improvements in alpha diversity and microbial composition, although these changes were not consistently beneficial. Moreover, associated changes in inflammatory and microbial translocation markers suggested unfavorable alterations. (4) Conclusions: Based on the evidence mapped, second-generation INSTIs may generally induce favorable changes in the gut microbiota. However, further research is needed to explore the clinical implications of these microbiota alterations, particularly in specific patient groups.

## 1. Introduction

### 1.1. Development and Characteristics of Antiretroviral Therapy

Since the introduction of antiretroviral therapy (ART) in 1987, the management of HIV has undergone a paradigm shift, transforming it from a fatal disease into a chronic but manageable condition. This advancement has significantly reduced HIV-related morbidity and mortality, improving the overall quality of life for people living with HIV (PLWH) [1,2]. Over the years, ART has evolved to include seven major drug classes. Each class targets distinct stages of the HIV replication cycle, thereby enhancing viral suppression and treatment outcomes [3]. The standard treatment for newly diagnosed ART-naïve patients usually consists of a combination of three active drugs, most commonly two nucleoside reverse transcriptase inhibitors (NRTIs) combined with either a second-generation INSTI, a boosted protease inhibitor (PI), or a non-nucleoside reverse transcriptase inhibitor (NNRTI) [4,5].

PIs were introduced in the 1990s as potent antiretroviral agents with a high genetic barrier to resistance [6,7]. They function by inhibiting HIV protease, preventing the cleavage of viral polyproteins, and thereby blocking the maturation of viral particles [8,9,10]. However, their use has been associated with serious metabolic and cardiovascular side effects and significant drug–drug interactions [6,7]. Nonetheless, PIs continue to play a crucial role in first-line therapy for select patients and remain an option for those who experience treatment failure with other ART drug classes [5,11,12].

Since the approval of the NNRTIs in 1996, this drug class has been a cornerstone of ART regimens [4,5,13]. First-generation NNRTIs, such as efavirenz, function by directly binding to and inhibiting reverse transcriptase, thereby preventing the enzyme from functioning [8,9,10]. Despite their effectiveness, these agents have notable limitations, including a low genetic barrier to resistance, which increases the risk of developing both drug-specific and cross-class resistance. In contrast, second-generation NNRTIs exhibit improved resistance profiles, lower toxicity, fewer drug–drug interactions, and enhanced synergistic effects when used in combination with other antiretroviral agents [14,15]. Although efavirenz was widely used for an extended period as a first-line treatment, its use has become more limited due to the emergence of neuropsychiatric side effects, the potential for the transmission of resistance mutations, and the emergence of novel INSTIs [16].

### 1.2. Integrase Strand Transfer Inhibitors

The first-generation INSTIs raltegravir and elvitegravir were introduced in 2007 and 2012, respectively, and provided effective viral suppression with minimal drug–drug interactions for raltegravir and good tolerability for both [2,17,18]. These agents work by preventing the integration of viral DNA into the host genome through direct binding to the integrase enzyme [8,9,10]. Compared to first-line NNRTI- and PI-based regimens, raltegravir- and elvitegravir-containing ART demonstrated good efficacy and safety, leading to their incorporation into treatment regimens for ART-naïve patients [19,20]. However, both agents exhibited a low genetic barrier to resistance, increasing the risk of cross-resistance and limiting subsequent treatment options within the INSTI class [16,21].

The second-generation INSTIs, including dolutegravir and bictegravir, were developed to address these limitations. Dolutegravir, introduced in 2013, has demonstrated superior efficacy when compared to darunavir/ritonavir and efavirenz/emtricitabine/tenofovir disoproxil fumarate, as well as non-inferiority to raltegravir [22,23,24]. Furthermore, its improved tolerability, fewer drug–drug interactions, and a significantly higher resistance barrier compared to first-generation INSTIs led the World Health Organization (WHO) in 2019 to recommend the transition from efavirenz-based regimens to dolutegravir-based therapy as the preferred first-line treatment for PLWH [16,25,26]. Bictegravir, a novel INSTI introduced in 2018, has also been introduced as part of a fixed-dose combination regimen [27]. It offered additional advantages such as suitability for rapid treatment initiation, no requirement for prior *HLA-B*5701* testing, compatibility with hepatitis B virus coinfection, and possibility of use in ART-experienced patients undergoing hemodialysis [28,29]. Additionally, cabotegravir, a long-acting INSTI, has been approved in combination with rilpivirine as a complete parenteral regimen for virally suppressed patients [30].

As the use of INSTIs continues to expand, their tolerability and side-effect profiles are becoming more clearly defined. Increasing evidence has raised concerns that second-generation INSTIs may be linked to observed side effects, such as adverse metabolic effects, including significant weight gain, as well as cardiovascular complications and other metabolic disturbances [26,29,31].

Currently, second-generation INSTIs are considered the preferred first-line therapies, being the most commonly administered ART and taken by millions of PLWH, whereas first-generation INSTIs, PIs, and NNRTIs receive varying levels of support from current treatment guidelines [4,5,16]. With accumulating clinical data, extensive research is ongoing to better characterize the differences between individual INSTIs, with a focus on their respective advantages and disadvantages in terms of efficacy, resistance, safety, and long-term outcomes [32,33,34,35].

### 1.3. The Role of Gut Microbiota in Chronic Inflammation Among PLWH

A crucial aspect of HIV infection is its impact on the human microbiome, particularly the composition of the gut microbiota. The gut microbiota comprises a diverse array of bacteria, viruses, and eukaryotic cells that maintain homeostasis with the human host [36,37]. The predominant bacterial phyla include *Firmicutes*, *Bacteroidetes*, *Actinobacteria*, *Proteobacteria*, *Fusobacteria*, and *Verrucomicrobia* [38,39,40]. The composition of the microbiota can be described using alpha diversity (defined by the evenness and richness of individual samples, measured by Shannon diversity and Chao1 index), beta diversity (defined by the variability of samples within a specific habitat), microbiota composition changes, and alterations in systemic inflammation or translocation markers [41]. The review that follows adheres to this framework.

During HIV infection, CD4+ T-cell depletion begins, significantly affecting the gut-associated lymphoid tissue (GALT). The compromised lymphoid system weakens the integrity of the intestinal barrier, allowing microbiota and associated metabolites to translocate into the systemic circulation, thereby triggering systemic inflammation. The translocation of diverse microbial antigens across the gut barrier activates the immune system, resulting in chronic immune activation and systemic inflammation [42,43]. Although findings on HIV-associated alterations in gut microbiota composition vary, a general trend toward reduced alpha diversity and shifts in beta diversity has been consistently reported [44]. An increase in *Proteobacteria* genera and a decrease in *Bacteroidetes* and several *Firmicutes*-associated genera is frequently reported. Additionally, *Ruminococcaceae* and *Prevotellaceae* have been linked to inflammatory properties and CD4+ T-cell counts [45,46,47]. Beyond HIV infection itself, other factors such as age, gender, geographical context, and comorbidities also contribute to the specific composition of the microbiota [48,49,50,51]. The previously observed changes in the gut microbiota results in dysbiosis and persistent systemic inflammation, which contributes to the development of noncommunicable diseases (NCDs), such as diabetes, metabolic syndrome, and cardiovascular diseases, and to an increased risk of mortality among PLWH [52,53].

Dysbiosis in the gut also contributes to significant alterations in the gut’s metabolomic profile, with one of the key markers being changes in short-chain fatty acid (SCFA) levels [54,55]. A commonly observed change is the alteration of propionate, which primarily exhibits anti-inflammatory properties. HIV infection disrupts the abundance of propionate-producing bacteria, particularly *Ruminococcaceae* and *Lachnospiraceae* from the *Firmicutes* phylum, resulting in decreased concentrations [54].

Beyond metabolic alterations, chronic inflammation in HIV infection is characterized by elevated levels of systemic inflammatory markers, including soluble cluster of differentiation 14 (sCD14), various interleukins, C-reactive protein (CRP), and intercellular adhesion molecule 1 (ICAM-1) [52,56,57,58]. Concurrently, microbial translocation markers such as lipopolysaccharides (LPS) and intestinal fatty acid-binding protein (I-FABP) are also altered, indicating increased permeability of the gut barrier and translocation of microbial components into the bloodstream [52,59].

Following HIV diagnosis, ART is initiated. While ART effectively suppresses viral replication and restores immune function, its impact on gut microbiota composition and the associated inflammatory and translocation markers remains an area of active investigation [60]. INSTIs have emerged as preferred first-line agents due to their high efficacy and tolerability [16,26]. However, understanding the differential effects of individual INSTIs on the gut microbiota is crucial, particularly in determining which agent offers the most favorable balance between efficacy and overall tolerability. The first-generation INSTIs remain a key focus, as their effects on the gut microbiota are often grouped with those of newer agents despite potential differences in their impact.

### 1.4. Aims of the Scoping Review

The objective of this scoping review was to systematically examine and update the current body of evidence regarding the effects of individual INSTIs on the gut microbiota, with an emphasis on identifying differences between agents and their potential clinical implications. This review builds upon our previous report, which explored changes in human gut microbiota following NNRTI- and INSTI-based therapies [61].

This review addresses two central research aims:To map alterations in the gut microbiota of PLWH receiving specific INSTI-based therapy in comparison to those undergoing NNRTI or PI-based regimens.To assess differences in gut microbiota composition between PLWH treated with dolutegravir-based regimens and those receiving bictegravir-based therapy.

## 2. Materials and Methods

This scoping review was conducted following the guidelines outlined in the PRISMA (Preferred Reporting Items for Systematic Reviews and Meta-Analyses) extension for scoping reviews [62]. Comprehensive literature searches were conducted in PubMed (21 April 2025), Web of Science (21 April 2025), and Embase (21 April 2025) from their respective inception dates, without applying restrictions on language or publication type. To ensure exhaustive search, a combination of Medical Subject Headings (MeSH) terms and free-text keywords was employed. The search approach was evaluated using seven preselected studies known to be relevant and was refined accordingly. Detailed information on the search methodology and the PRISMA extension checklist for scoping reviews are available in Appendix A, respectively. Additionally, reference lists from relevant systematic reviews were manually examined to identify any further eligible studies.

The study selection process comprised three stages: (1) removal of duplicates, (2) screening of titles and abstracts, and (3) full-text review. Selection criteria were based on predefined inclusion and exclusion parameters (Table 1). The first reviewer completed all stages, with the final stage reviewed for accuracy by a second reviewer. Duplicate detection and data management was performed using Rayyan and Excel (version 2013). Any disagreements during the selection process were resolved through discussion.

Relevant data were extracted by one reviewer and independently verified by another to ensure accuracy. The extracted information included the following: study title, authors, year of publication, country of origin, study design, number of participants, participant characteristics, details of the intervention, methods of stool sample data collection, duration of follow-up, and key outcomes (including alpha and beta diversity indices, microbiota composition, and changes in markers of bacterial translocation or systemic inflammation).

The study selection process is depicted in Figure 1, while the findings from the included studies are summarized in Table 2 and Figure 2. These summaries provide a comprehensive update of the current research landscape and highlight potential gaps in knowledge. Publications that detail the effects of specific INSTI therapies but were excluded due to unsuitable publication types were incorporated into the Discussion section to provide a broader perspective. As this is a scoping review, a formal assessment of the risk of bias was not conducted.

## 3. Results

### 3.1. Study Selection Process

A total of 13,140 articles were retrieved from the PubMed, Web of Science, and Embase databases. After removing 3852 duplicates, 9081 records were excluded through title and abstract screening based on relevance, language, publication type, and text availability. Of the 224 full-text articles assessed for eligibility, 201 were excluded for the following reasons: (1) absence of detailed information regarding the effects of ART on the gut microbiota or on markers of inflammation; (2) inclusion of ART-naïve participants; (3) focus on non-INSTI-based ART regimens; and (4) failure to specify the particular INSTI administered or the individual effects of specific INSTIs. The screening and selection process is illustrated in Figure 1. Ultimately, six studies met the inclusion criteria and were included in the final analysis.
Figure 1Prisma flowchart on the study selection process.
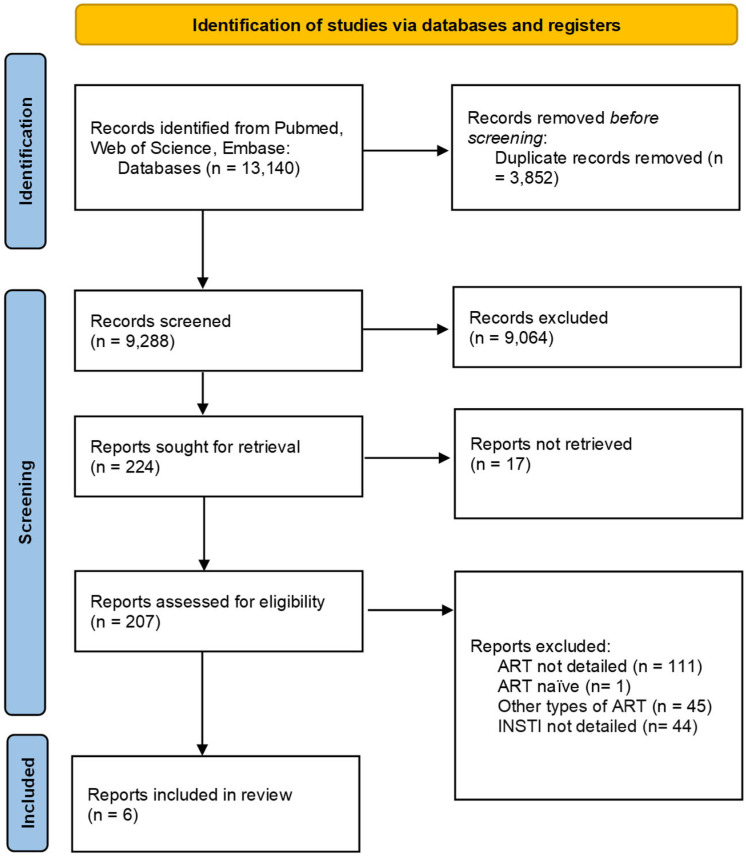


### 3.2. Specific INSTI Related Changes in the Gut Microbiota

A total of six studies were included in the final analysis. These consisted of three randomized controlled trials and one cross-sectional study comparing raltegravir-based regimens with NNRTI- and/or PI-based therapies; one non-randomized trial assessing the effects of dolutegravir and bictegravir in comparison with NNRTI-based regimens; and one comparative study investigating the effects of dolutegravir versus an NNRTI-based regimen.

Hanttu et al. examined gut microbiota composition in patients receiving efavirenz or a PI compared to those who switched from efavirenz- or PI-based therapy to raltegravir as well as to HIV-seronegative controls. Serum levels of I-FABP, LPS, calprotectin, and vitamin D were also measured, with a follow-up period of 24 weeks [63]. At baseline, the efavirenz-treated group exhibited lower alpha diversity (Chao-1 index) compared to HIV-seronegative individuals. Beta diversity was significantly altered in both PI- and efavirenz-treated groups relative to HIV-seronegative individuals, with distinct microbial composition differences among the three subgroups [63]. Following the switch to raltegravir, alpha diversity (Chao-1 and Shannon indices) significantly increased in the efavirenz-to-raltegravir group, approaching levels seen in HIV-seronegative individuals [63]. Microbial composition showed a trend towards increased *Prevotella-9* and decreased *Phascolarctobacterium* and *Bacteroides* after switching from PI or efavirenz to raltegravir, though these changes could not be directly attributed to either prior regimen. Biomarker analysis indicated higher baseline I-FABP levels in the efavirenz-treated group, but switching to raltegravir did not significantly affect either biomarker levels [63].

Villanueva-Millán et al. conducted a cross-sectional study involving PLWH receiving raltegravir-based (n = 8), NNRTI-based (n = 22), or PI-based (n = 15) ART, alongside ART-naïve PLWH (n = 5) and seronegative individuals (n = 21) [64]. Alpha diversity in the raltegravir-treated group was comparable to that of healthy controls and significantly higher than in ART-naïve individuals. In terms of microbiota composition, *Bacteroidetes* and *Firmicutes* were the dominant phyla across the entire study population. Raltegravir treatment was associated with an increased abundance of the *δ-Proteobacteria* class; the *Desulfovibrionales* and *Selenomonadales* orders; the *Desulfovibrionaceae* and *Lachnospiraceae* families; the *Desulfovibrio* genus; as well as *Blautia* sp. *3* and *Parabacteroides merdae* species and decreased levels of the *Clostridiales* order [64]. Furthermore, compared to PI-based regimens, raltegravir was linked to a lesser reduction in bacterial species richness. Regarding translocation markers, sCD14 levels in the raltegravir group were similar to those observed in healthy controls, unlike in other ART-treated groups. However, levels of ICAM-1 were elevated [64].

El Kamari et al. conducted a randomized controlled trial in which ART-naïve PLWH were assigned to receive either raltegravir- (n = 82), darunavir/ritonavir- (n = 82), or atazanavir/ritonavir-based (n = 67) ART regimens. The study assessed markers of gut integrity, including lipopolysaccharide-binding protein (LBP), zonulin, I-FABP, and ileal bile acid-binding protein (I-BABP), over a 96-week period [65]. Participants in the raltegravir arm exhibited significantly higher zonulin levels compared to those in the other treatment groups, while increases in I-BABP and LBP levels were observed but did not reach statistical significance. I-FABP levels—associated with increases in BMI, as well as visceral, subcutaneous, and total adipose tissue—showed a consistent elevation across all treatment arms and remained comparable among the different regimens [65].

Kelesidis et al. conducted a study similar to that of El Kamari et al., enrolling ART-naïve PLWH and assigning them to receive raltegravir- (n = 106), darunavir/ritonavir- (n = 113), or atazanavir/ritonavir-based (n = 109) ART regimens. The study evaluated a panel of inflammatory and coagulation markers, including CRP, IL-6, glycoprotein acetylation (GlycA), D-dimer, sCD14, soluble CD163 (sCD163), and soluble interleukin-2 receptor (sIL-2r), measured at 24 or 48 weeks and at 96 weeks [66]. In the raltegravir treatment arm, reductions in hs-CRP, IL-6, GlycA, sCD14, sCD163, and sIL-2r levels were observed. However, changes in inflammatory markers did not differ significantly between the raltegravir and PI-based treatment arms [66].

Narayanan et al. conducted a study involving 80 HIV-seronegative individuals and 69 PLWH receiving long-term ART with PI-, NNRTI-, or INSTI-based regimens. Among individuals on long-term treatment, the bictegravir-based ART subgroup exhibited a higher abundance of *Bifidobacterium*, *Anaerostipes*, *Butyricimonas*, and *Butyricicoccus*, while *Faecalibacterium* and *Ruminococcus gauvreauii* were enriched in those receiving dolutegravir. In contrast, *Megasphaera* was more abundant in NNRTI-treated individuals [67]. Further analysis focused on patients receiving dolutegravir. Those with a high BMI exhibited increased abundances of *Bifidobacterium*, *Dorea*, and *Streptococcus*, whereas those with a low BMI showed higher levels of *Bacteroides* and *Escherichia-Shigella*. Age-related differences were also observed: younger PLWH on dolutegravir had lower alpha diversity, with an enrichment of *Lachnospira* and *Eggerthella*, while older PLWH exhibited higher alpha diversity with increased *Coprococcus* and *Dorea* levels. Beta diversity also differed significantly between these age groups [67]. Additionally, long-term dolutegravir-treated patients had higher alpha diversity and an increased abundance of *Succinivibrio* compared to those on short-term treatment. CD4 cell count was also associated with microbial composition: patients with high CD4 counts had greater species richness, with elevated levels of *Dialister*, *Ruminococcus*, and *Agathobacter*, whereas those with lower CD4 counts exhibited enrichment of *Fusobacterium* and *Ruminococcus gnavus* [67].

Roux et al. analyzed inflammatory markers in PLWH who switched from efavirenz to a dolutegravir-based regimen (n = 9) and compared them to HIV-seronegative controls (n = 20). Inflammatory markers were assessed at baseline and after six months of dolutegravir treatment. Based on CRP levels, with a threshold of 5 mg/L, participants were stratified into two subgroups for further analysis [68]. At six months, all dolutegravir-treated patients exhibited significantly elevated levels of leucine, MIP-1α, sCD40L, and RANTES compared to healthy controls. However, after excluding patients with persistently elevated CRP (n = 4/9), only MIP-1α remained significantly increased [68]. Regarding metabolic profiles, at baseline, the efavirenz-treated group had higher levels of AMP, 1,7-dimethylxanthine, formic acid, glucose, and glycolic acid, while acetoacetic acid, creatinine, lactic acid, myo-inositol, and urea were reduced compared to seronegative individuals. Following the switch to dolutegravir, only 3-hydroxybutyric acid showed a significant increase, whereas other metabolites exhibited less pronounced changes [68]. In terms of inflammatory markers, the efavirenz-treated group demonstrated elevated G-CSF, GM-CSF, and PDGF-BB levels compared to healthy controls. After six months on dolutegravir, G-CSF and MIP-1α remained elevated, while IL-6, GM-CSF, and PDGF-BB decreased to levels comparable to seronegative individuals. Additionally, platelet activation markers, including sCD40L and RANTES, were elevated during efavirenz treatment but normalized following the switch to dolutegravir [68].

The study results are summarized in Table 2, while the specific changes induced by dolutegravir, bictegravir, and raltegravir are presented in Figure 2, along with the taxonomy of the bacteria discussed in this article.
Figure 2The taxonomy of the bacteria discussed in this article, with a particular focus on the specific changes induced by dolutegravir, bictegravir, and raltegravir [40,63,64,65,66,67,68,69,70,71,72,73].
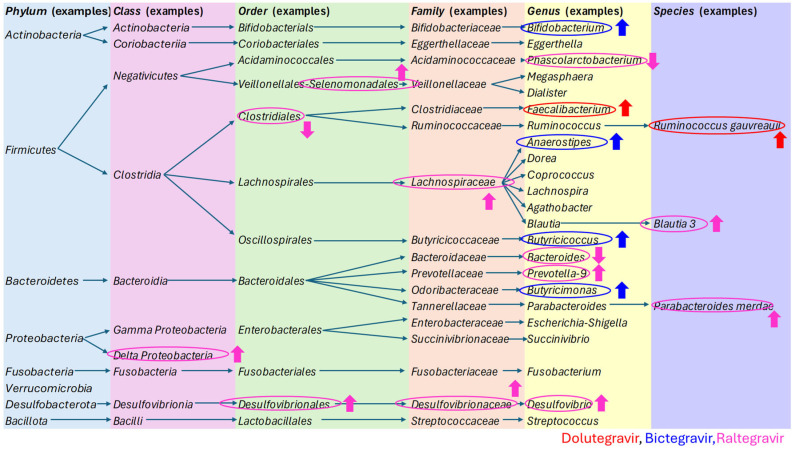

ijms-26-06366-t002_Table 2Table 2Raltegravir-, dolutegravir-, and bictegravir-specific changes on the gut microbiota in PLWH.Study TitleAuthorPublication YearParticipantsINSTI-Mediated Alpha Diversity ChangesINSTI-Mediated Beta Diversity ChangesINSTI-Mediated Changes in Microbiome CompositionStool Sample AnalysisINSTI-Mediated Change on Translocation or Inflammation MarkersGut microbiota alterations after switching from a protease inhibitor or efavirenz to raltegravir in a randomized, controlled study [63]Hanttu et al.2023PLWH on either efavirenz or PI (n = 41) vs. 24 weeks after switch to raltegravir (n = 19) vs. negative controls (n = 10)Raltegravir treatment approximated the alpha diversity of HIV-seronegative individuals-Raltegravir:
*Prevotella-9* ↑*Phascolarctobacterium*, *Bacteroides* ↓
16S rRNA sequencingNo changesDifferential effects of antiretrovirals on microbial translocation and gut microbiota composition of HIV-infected patients [64]Villanueva-Millán et al.2017PLWH on either NNRI (n = 22) or PI (n = 15) or raltegravir (n = 8) vs. ART-naïve (n = 5) and negative controls (n = 21)Raltegravir treatment approximated the alpha diversity of HIV-seronegative individuals-Raltegravir:
*δ-Proteobacteria* class, *Desulfovibrionales* and *Selenomonadales* orders, *Desulfovibrionaceae* and *Lachnospiraceae* families, and *Desulfovibrio* genus, *Blautia* sp. *3*, *Parabacteroides merdae* species ↑*Clostridiales* ↓
16S rDNA pyrosequencingRaltegravir
sCD14 levels were similar to controlsICAM ↑
Lower Pretreatment Gut Integrity Is Independently Associated With Fat Gain on Antiretroviral Therapy [65]El Kamari et al.2018PLWH randomized to raltegravir (n = 82), darunavir/ritonavir (n = 82), and atazanavir/ritonavir (n = 67) therapies----Raltegravir
Zonulin, I-FABP ↑I-FABP levels did not significantly differ between raltegravir and PI-based treatment arms
Changes in Inflammation and Immune Activation With Atazanavir-, Raltegravir-, Darunavir-Based Initial Antiviral Therapy: ACTG 5260s [66]Kelesidis et al.2015PLWH randomized to raltegravir (n = 106), darunavir/ritonavir (n = 113), and atazanavir/ritonavir (n = 109) therapies----Raltegravir
hs-CRP, IL-6, GLycA, sCD14, sCD163, sIL-2r ↓No significant difference between raltegravir and PI-based treatment arms
Exploring the interplay between antiretroviral therapy and the gut-oral microbiome axis in people living with HIV [67]Narayanan et al.2023[INSTI (bictegravir vs. dolutegravir) vs. NNRTI vs. PI] (n = 69) vs. [negative controls] (n = 80)Dolutegravir:
Younger age: lowerOlder age: higherLong-term treatment: higherHigh CD4: higher
Dolutegravir:
Differed between younger and older dolutegravir-treated patientsBictegravir:
*Bifidobacterium*, *Anaerostipes*, *Butyricimonas*, *Butyricicoccus* ↑
Dolutegravir:
*Faecalibacterium*, *Ruminococcus gauvreauii* ↑High BMI: *Bifidobacterium*, *Dorea*, *Streptococcus* ↑Low BMI: *Bacteroides*, *Escherichia-Shigella* ↑Younger age: *Lachnospira*, *Eggerthella* ↑Older age: *Coprococcus Dorea* ↑Long-term treatment: *Succinivibrio* ↑High CD4: *Dialister*, *Ruminococcus*, *Agathobacter* ↑Lower CD4: *Fusobacterium*, *Ruminococcus gnavus* ↑
16S rRNA sequencing-Comparative Effects of Efavirenz and Dolutegravir on Metabolomic and Inflammatory Profiles, and Platelet Activation of People Living with HIV: A Pilot Study [68]Roux et al.2024Efavirenz (n = 9) vs. 6 months after dolutegravir switch (n = 9) vs. negative controls (n = 20)---16S rRNA sequencingDolutegravir:
Leucine, MIP-1α, sCD40L, RANTES ↑
Dolutegravir with normalizing CRP levels:
MIP-1α ↑3-hydroxybutyric ↑IL-6, GM-CSF, PDGF-BB ↓sCD40, RANTES ↓


## 4. Discussion

In the present scoping review, our objective was to synthesize the existing knowledge on gut microbiota changes associated with specific INSTI-based therapies. This review builds upon our previous scoping review, which detailed microbiota alterations related to INSTI- and NNRTI-based therapies. The previous review summarized that INSTI-based regimens may improve alpha diversity, approaching levels observed in healthy controls, while beta diversity remains relatively unchanged. The findings also suggested that INSTI therapies enhance microbiome composition and may contribute to a reduction in inflammatory markers [61]. The present findings reinforce these observations, with dolutegravir demonstrating beneficial effects on both alpha and beta diversity as well as on overall gut microbiota composition. However, factors such as younger age, persistently low CD4+ T-cell counts, and extreme BMI were associated with unfavorable microbiota shifts. Available evidence also suggested that bictegravir promotes favorable gut microbiota alterations. Furthermore, raltegravir may be linked to improvements in alpha diversity and composition, though these changes were not universally beneficial. Notably, changes in gut integrity and inflammatory markers associated with raltegravir did not differ significantly from those observed with PI-based regimens [63,64,65,66,67,68]. While the current review builds on prior systematic findings, it provides a more detailed, agent-specific analysis of individual INSTIs.

In more detail, among the current studies, the most comprehensive data from second-line INSTIs was available for dolutegravir. Dolutegravir treatment was linked to improved alpha diversity in most subgroups except for younger patients (<40 years). Significant differences in beta diversity were observed exclusively in relation to age. Moreover, dolutegravir use was linked to the enrichment of *Faecalibacterium* and *Ruminococcus gauvreauii* [67,68]. *Faecalibacterium* has strong anti-inflammatory properties, and its reduced levels are associated with inflammatory bowel disease, diabetes mellitus, and obesity [74]. *Ruminococcus gauvreauii* is indicative of adequate thiamin levels, essential for immune regulation, and its depletion may suggest a shift toward a proinflammatory immune profile [75]. The study further emphasized the role of age, BMI, CD4+ T-cell count, and treatment duration in shaping microbiota composition, alongside the effects of dolutegravir treatment [67].

Narayana et al. observed that age significantly influenced both alpha and beta diversity. In the literature, De la Cuesta-Zuluaga et al. reported that alpha diversity increases until approximately 40 years before plateauing [76], whereas Badal et al. observed a subsequent rise in the oldest individuals [77]. The literature data also suggest that aging induces shifts in beta diversity, marked by an increase in proinflammatory facultative anaerobes and a decline in obligate anaerobes, which are essential for gut homeostasis and immune regulation [78]. Additionally, the enrichment of *Lachnospira* and *Eggerthella* among younger PLWH on dolutegravir suggests metabolic disturbances linked to obesity, diabetes, chronic kidney disease, and liver disorders [79]. In contrast, among older PLWH on dolutegravir, an increase in the beneficial butyrate-producing genus *Coprococcus* was observed alongside elevated *Dorea* levels. *Dorea* is associated with insulin production and fasting blood glucose levels, potentially contributing to type 2 diabetes mellitus and weight gain [80,81,82].

Treatment duration and CD4 cell count were also significant determinants of gut microbiota composition in PLWH receiving dolutegravir-based treatment. Consistent with previous findings, lower CD4 cell counts were associated with an increased abundance of *Fusobacterium*, while long-term treatment was linked to *Succinivibrio* enrichment, both of which are implicated in microbial alterations seen in immunological nonresponders and responders, respectively [67,83,84]. Additionally, the study reported elevated *Ruminococcus gnavus* levels in PLWH with low CD4 cell counts on dolutegravir treatment. This bacterium plays a complex role by producing SCFAs while simultaneously disrupting the gut mucus barrier. It has also been associated with metabolic syndrome, diabetes, and obesity [85].

This study further analyzed microbiota composition in PLWH receiving dolutegravir therapy in relation to BMI. High BMI was associated with *Bifidobacterium* enrichment [67]. *Bifidobacterium* is commonly associated with markers of good health. It functions as a primary degrader in the gut microbiome, contributes to the maintenance of gut homeostasis and has been linked to protective effects against weight gain [86,87,88,89]. However, elevated *Dorea* levels were also observed, suggesting a microbiome alteration that could facilitate weight gain and metabolic disorders [67,86,87,89]. Conversely, low BMI was linked to increased levels of opportunistic pathogens such as *Escherichia-Shigella* [90]. Surprisingly, elevated levels of *Bacteroides* were also observed, with *Bacteroides* promoting immune modulation through invariant natural killer T cells and IL-4 production [91]. Interestingly, a preprint study by Blazquez-Bondia et al. examined the human gut microbiome composition of PLWH initiating either DTG (n = 46) or darunavir/ritonavir (n = 42) therapy and reported an increased abundance of multiple *Bifidobacterium* species in the DTG treatment arm. However, these microbial changes were not associated with alterations of body weight [92].

Regarding inflammatory and microbial translocation biomarkers, Roux et al. demonstrated a favorable immunological shift following the transition from efavirenz to dolutegravir. Specifically, platelet activation markers such as RANTES and sCD40, along with proinflammatory markers including IL-6, GM-CSF, and PDGF-BB, significantly decreased after dolutegravir initiation. This reduction suggests a decline in chronic immune activation, which is associated with the development of NCD such as atherosclerosis and cardiovascular diseases [68,93]. Additionally, an elevation in 3-hydroxybutyrate was observed, a key energy source essential for gut homeostasis [68,94]. However, an increase in MIP-1α, which plays a role in myeloid cell recruitment, was also noted, while the inflammatory marker G-CSF remained unchanged [68,95].

With respect to bictegravir therapy, compositional changes were observed in a current study by Narayanan et al., including the enrichment of *Bifidobacterium*, *Anaerostipes*, *Butyricimonas*, and *Butyricicoccus* [67]. Evidence suggests that *Bifidobacterium*, *Anaerostipes*, *Butyricimonas*, and *Butyricicoccus* are beneficial due to their ability to produce the SCFA butyrate, which plays a crucial role in gut health [96,97,98,99,100]. In vitro models further suggested that *Butyricimonas* may play a protective role against diabetes mellitus and metabolic disorders [101]. Although an in vitro study by Rubio-García et al. also demonstrated antimicrobial effects of bictegravir against *Enterococcus* spp., pharmacokinetic studies suggest that bictegravir does not reach sufficient concentrations in the gut microbiota to significantly impact their abundance [102,103].

Although raltegravir is now primarily considered a second-line treatment, it is frequently assessed alongside second-generation INSTI therapies in studies examining gut microbiota changes following INSTI initiation. However, potential differences in their effects should be considered. The studies included in the present review suggested that raltegravir therapy enhances alpha diversity, approaching levels observed in healthy individuals. This effect was observed almost uniformly alongside INSTI therapies [64,104,105].

Regarding bacterial composition, the study by Hanttu et al. observed increased abundances of *Prevotella-9* and a decrease in *Phascolarctobacterium* and *Bacteroides* levels [63]. In general, *Prevotella*, a proinflammatory taxon, is associated with the microbiota composition of MSM and has been linked to both improved glucose metabolism and insulin resistance [48,106,107]. Additionally, Hishiya et al. found that *Prevotella-9* was specifically more abundant in patients with higher CD4 counts, while elevated fecal succinic acid in low CD4 patients negatively impacted its levels [108]. Furthermore, *Phascolarctobacterium* is linked to inflammatory cytokines, with its abundance positively correlating with weight loss and its reduction associated with inflammatory bowel disease [109,110,111]. Villanueva-Millán et al. reported increases in microbial taxa traditionally considered markers of dysbiosis, including the genus *Desulfovibrio*, the class δ-*Proteobacteria*, and members of the family *Lachnospiraceae*, all of which are implicated in metabolic disorders and atherosclerosis [64,112,113]. The observed reduction in the order *Clostridiales*, which plays a key role in maintaining gut homeostasis and overall gastrointestinal function, is also conflicting [64,114]. Conversely, increases in bacteria known for their production of SCFAs—such as those from the order *Selenomonadales*, family *Lachnospiraceae*, and species *Blautia* and *Parabacteroides merdae*—may indicate protective microbial shifts associated with RAL-based therapy [64,79,115,116]. Pharmacokinetic studies by Patterson et al. and Thompson et al. offer a potential mechanistic explanation, demonstrating that raltegravir (RAL) accumulates at high concentrations across multiple sites within the human gastrointestinal tract. This accumulation may underlie its distinct and diverse effects on gut microbiota composition [117,118].

Regarding alterations in gut integrity and systemic inflammatory markers, both the study by Hanttu et al. and Kelesidis et al. reported a favorable reduction in sCD14 levels—a biomarker of monocyte activation and a well-established predictor of HIV-associated morbidity and mortality [63,66,119]. In contrast, the study by Kelesidis et al. demonstrated non-consistent reductions of inflammatory markers after RAL and the changes did not differ significantly from the PI treatment arm. These alterations are indicative of a sustained proinflammatory state that may contribute to ongoing immune activation despite virologic suppression [66,120,121]. On the other hand, elevated zonulin levels—an established regulator of intestinal permeability—were particularly noted in the RAL treatment group in the study conducted by El Kamari et al. Previous research has suggested that zonulin production is suppressed in untreated HIV infection due to epithelial damage in the intestinal barrier. This suppression may be partially reversed upon initiation of ART, resulting in increased zonulin levels and consequently enhanced gut permeability [65,122].

Weight gain and the associated risk of NCDs among PLWH represent a significant challenge in modern HIV therapy. The factors contributing to weight gain have been extensively analyzed, with a particular focus on the potential role of INSTI-based therapies [29,123,124,125]. The present scoping review did not aim to synthesize evidence related to weight gain, and the existing data on microbiome alterations in association with weight changes remain inconclusive. Nevertheless, existing studies suggest that dolutegravir therapy may offer some protection against weight gain except for individuals with a high BMI, with a low CD4 count, or at certain age thresholds. In the case of bictegravir, the available evidence remains limited, and a protective effect against metabolic dysregulation has been hypothesized. However, a conference presentation by Pinto-Cardoso et al. reported significant weight gain following a switch from efavirenz to bictegravir-based therapy among treatment-experienced PLWH [67,126].

Regarding the effects of RAL therapies El Kamari et al. demonstrated that RAL-based regimens apply effects on gut integrity markers that are comparable to those observed with PI-based therapies. Both treatment modalities were associated with elevated and sustained levels of I-FABP. Increased I-FABP levels have been linked to weight gain, and it has been hypothesized that this may be mediated through the stimulation of hepatic triglyceride synthesis and subsequent adipose tissue accumulation [65,127]. Importantly, the study also emphasized the critical influence of baseline gut function on BMI and fat distribution patterns [65]. Overall, this complex phenomenon appears to be multifactorial and may be partly explained by the “return to health” effect, implying a limited contribution from direct ART-associated metabolic adverse effects [128,129,130,131,132].

To our knowledge, this is the first scoping review to assess specific changes in gut microbiota and associated inflammatory markers in response to dolutegravir-, bictegravir-, and raltegravir-based ART treatment. However, our study has several limitations. First, the included studies examined a relatively small patient population, which may limit the generalizability of the findings. Second, the microbiota analysis was mostly based on the 16S rRNA sequencing method. Evidence suggests that 16S rRNA sequencing primarily predicts specific genes based on amplified regions rather than directly sequencing the entire rRNA gene. Additionally, this method may not detect subtle alterations in microbial composition [133]. Moreover, Roux et al. applied a 10% significance level, which may have further influenced the results and their interpretation. In spite of these limitations, this scoping review is first of its kind aligning the available current scientific literature on the effects of the most commonly used INSTIs on the gut microbiota.

## 5. Conclusions

To summarize, among the studies reviewed, the most substantial body of evidence of second-line INSTIs was found for dolutegravir, which demonstrated beneficial effects on both alpha and beta diversity, as well as on the overall composition of the gut microbiota. However, factors such as younger age, persistently lower CD4+ T-cell counts, and extreme BMI were associated with unfavorable shifts in microbiota composition. Although data on bictegravir was more limited, the available evidence suggested that bictegravir also induces favorable alterations in the gut microbiota. Furthermore, raltegravir was linked to improvements in alpha diversity and composition, although these changes were not universally beneficial. Moreover, associated changes in inflammatory and microbial translocation markers suggested unfavorable alterations.

Additional studies are required to better define the impact of first-line ART therapies on the gut microbiota and to investigate its role in disease progression and inflammation. Gaining deeper insight into these relationships could aid clinicians in choosing treatments that reduce gut dysbiosis-related chronic inflammation while identifying the key factors driving microbiota alterations.

## Figures and Tables

**Table 1 ijms-26-06366-t001:** Inclusion and exclusion criteria.

Inclusion criteria			
Studies published in English were included if they belonged to one of the following categories: case-control studies, cohort studies, single-arm trials, ort randomized controlled trials.			
	**Population**	**Intervention**	**Comparator**
PICO 1	Adults diagnosed with HIV infection confirmed by serological tests	PLWH receiving INSTI-based therapy	PLWH receiving NNRTI/PI-based therapy
PICO 2	Adults diagnosed with HIV infection confirmed by serological tests	PLWH receiving dolutegravir-based therapy	PLWH receiving bictegravir-based therapy
Exclusion criteria			
The following types of publications were excluded: reviews, case series, case reports, clinical guidelines, conference abstracts, letters, preprints, and editorials.			

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
