# Peer review of "Mapping the Gut Microbiota Composition in the Context of Raltegravir, Dolutegravir, and Bictegravir—A Scoping Review"

_ijms, 2025, doi:10.3390/ijms26136366_

Round 1
Reviewer 1 Report (New Reviewer)
Comments and Suggestions for Authors
Observations:
- Why include reviews in search protocol?
- Six studies in a comprehensive review are not enough to conclude on the overal efefct on microbiome changes induced by INSTIs.
- No metaregression to show association between independent variables and microbiome changes
English fine
Author Response
Comment 1: Why include reviews in search protocol?
Response 1: Thank you for your feedback. Review articles were excluded during the second phase of the screening process (screening of titles and abstracts). Their listing under the inclusion criteria in the original figure was an oversight. This error has been corrected, and the figure has been appropriately updated.
Comment 2: Six studies in a comprehensive review are not enough to conclude on the overal efefct on microbiome changes induced by INSTIs.
Response 2: Thank you for this valuable remark. We have carefully revised the ‘Discussion’ and ‘Conclusion’ sections to enhance the clarity of our focus on mapping the existing evidence, while refraining from drawing definitive conclusions.
Comment 3: No metaregression to show association between independent variables and microbiome changes
Response 3: Thank you for this observation. As this study is a scoping review rather than a systematic review, a formal assessment of the risk of bias was not conducted, in accordance with established methodological guidance for scoping reviews. This is stated in the ‘Materials and Methods’ section.
Additionally, we have refined the language throughout the manuscript to improve clarity and readability.
Reviewer 2 Report (New Reviewer)
Comments and Suggestions for Authors
1. The title of the manuscript is not related to the aim of the study:
The title mentions good microbiota, and the aim To map changes in the gut microbiota and To assess differences in the composition of the gut microbiota.
2. Item 3.1 Study selection process needs to be removed or not as detailed.
3. Have the outcomes of patients taking the dosage forms under consideration compared with a control group of patients.
Author Response
Comment 1: The title of the manuscript is not related to the aim of the study: The title mentions good microbiota, and the aim To map changes in the gut microbiota and To assess differences in the composition of the gut microbiota.
Response 1: Thank you for this remark. We have revised the title to more accurately reflect the objective of our scoping review.
Comment 2: Item 3.1 Study selection process needs to be removed or not as detailed.
Response 2: Thank you for this suggestion. We have shortened the section.
Comment 3: Have the outcomes of patients taking the dosage forms under consideration compared with a control group of patients.
Response 3: Thank you for this suggestion. However, we are uncertain whether we have fully understood your question as intended. We did not identify any additional studies that specifically analysed the effects of raltegravir, bictegravir, or dolutegravir in comparison either to healthy controls or to antiretroviral therapy-naïve individuals living with HIV. We made every effort to comprehensively map all available data regarding the individual effects of these integrase strand transfer inhibitors. Studies that did not meet the inclusion criteria for the results section have nevertheless been incorporated into the discussion to provide broader contextual and background information.
Round 2
Reviewer 2 Report (New Reviewer)
Comments and Suggestions for Authors
The authors have taken into account the recommendations.
This manuscript is a resubmission of an earlier submission. The following is a list of the peer review reports and author responses from that submission.
Round 1
Reviewer 1 Report
Comments and Suggestions for Authors
The manuscript "Update on the Effects of Raltegravir, Dolutegravir and Bicte- 2 gravir on the Gut Microbiota" is well structured and written. I recommend it for publication after major changes. Some observations are:
- Although the authors have a valid argumentation through out the manuscript, the central research questions are answered only with three manuscripts, which is not enough for a strong review.
- There are many abbreviations in some sections and is hard to follow, please consider decrease the number of abbreviations.
- Graphical abstract doesn't need figure number.
Author Response
Comment 1: Although the authors have a valid argumentation through out the manuscript, the central research questions are answered only with three manuscripts, which is not enough for a strong review.
Response 1: Thank you for this important remark. In our previous scoping review, we mapped the available evidence on the effects of integrase strand transfer inhibitors (INSTIs) and non-nucleoside reverse transcriptase inhibitors (NNRTIs), which also included findings related to the effects of INSTIs. We recognized that a more focused approach—mapping the evidence for each individual INSTI—was a relatively unexplored area. However, we were able to build upon the evidence gathered in the previous review and incorporate new findings through a more targeted search. This approach allowed us to identify more precise evidence gaps that could inform future clinical trials. From a methodological perspective, as this review follows a scoping review framework, the inclusion of three manuscripts is sufficient to map the existing evidence. Given the nature of a scoping review, further statistical comparisons were not conducted, as the number of included studies does not impact on the primary objective of synthesizing and summarizing the available literature.
Comment 2: There are many abbreviations in some sections and is hard to follow, please consider decrease the number of abbreviations.
Response 2: Thank you for this insightful remark. We tried to rationalize the use of abbreviations, omitting those that are unnecessary. However, we have retained commonly recognized abbreviations and those for inflammatory markers, as they are more widely understood in their abbreviated form than in their full names.
Comment 3: Graphical abstract doesn't need figure number.
Response 3: Thank you for the comment. We have corrected the text accordingly.
Reviewer 2 Report
Comments and Suggestions for Authors
This work synthesizes findings from three authors—Hanttu, Narayanan, and Roux—regarding the effects of second-generation integrase strand transfer inhibitors (INSTIs) on gut microbiota. The gut microbiome has recently gained significant attention, particularly for its potential role in people living with HIV. However, the analysis of these three publications does not provide a definitive conclusion on whether antiretrovirals (raltegravir, dolutegravir, and bictegravir) contribute to weight gain. Additionally, the authors designed a methodology to analyze existing literature but only identified three relevant publications. They do not discuss why other studies were excluded from their selection criteria. A broader review of the literature, including animal studies available in databases such as Google Scholar, should be considered to provide a more comprehensive understanding of the topic.
Author Response
Comment 1: This work synthesizes findings from three authors—Hanttu, Narayanan, and Roux—regarding the effects of second-generation integrase strand transfer inhibitors (INSTIs) on gut microbiota. The gut microbiome has recently gained significant attention, particularly for its potential role in people living with HIV. However, the analysis of these three publications does not provide a definitive conclusion on whether antiretrovirals (raltegravir, dolutegravir, and bictegravir) contribute to weight gain.
Response 1: Thank you for this important insight. The current scoping review focused exclusively on microbiome alterations in relation to specific INSTI therapies. However, we acknowledge the significance of this issue. Therefore, a brief discussion on the available evidence regarding the relationship between microbiome alterations and weight change for each INSTI agent has been included in the Discussion section. Nonetheless, the existing evidence is limited and remains inconclusive.
Comment 2: Additionally, the authors designed a methodology to analyze existing literature but only identified three relevant publications. They do not discuss why other studies were excluded from their selection criteria.
Response 2: Thank you for the opportunity to provide further clarification. Our central objective was to systematically map the evidence on the effects of each INSTI—bictegravir, dolutegravir, and raltegravir—on the gut microbiota. Following a comprehensive literature search across three relevant databases, we initially identified 12,857 records. The exclusion process was conducted in multiple stages. The first criterion was the removal of duplicate records. The second step involved title and abstract screening, where we excluded studies addressing irrelevant topics, those published in languages other than English, and those falling under unsuitable publication types such as case series, case reports, clinical guidelines, conference abstracts, letters, preprints, and editorials. The third exclusion criterion was the unavailability of full-text articles. Finally, in the full-text review stage, we excluded studies that did not focus on the relevant topics, lacked detailed information on antiretroviral therapy (ART) or INSTI specification, included ART-naïve patients, or primarily investigated other types of ART. We believe that this process is thoroughly detailed in Figure 1 and Section 3.1: Study Selection Process.
Comment 3: A broader review of the literature, including animal studies available in databases such as Google Scholar, should be considered to provide a more comprehensive understanding of the topic.
Response 3: Thank you for your comment. We aimed to adhere to a strict protocol, as this research field is relatively narrow, with limited available evidence. This approach was intended to minimize potential biases. However, we incorporated in vitro studies in the Discussion section to provide a broader perspective. Regarding animal studies, the risk of bias is considerably higher, making their findings less generalizable and, therefore, less suitable for inclusion in our analysis. We acknowledge that further research is needed to enhance our understanding of this topic.
Round 2
Reviewer 1 Report
Comments and Suggestions for Authors
I recommend the manuscript for publication, in the present form.